# Investigating the Impact of Sample Preparation on Mass Spectrometry-Based Drug-To-Antibody Ratio Determination for Cysteine- and Lysine-Linked Antibody–Drug Conjugates

**DOI:** 10.3390/antib9030046

**Published:** 2020-09-08

**Authors:** Malin Källsten, Rafael Hartmann, Lucia Kovac, Fredrik Lehmann, Sara Bergström Lind, Jonas Bergquist

**Affiliations:** 1Department of Chemistry-BMC, Uppsala University, S-75124 Uppsala, Sweden; sara.lind@kemi.uu.se; 2Recipharm OT Chemistry AB, S-75450 Uppsala, Sweden; lucia.kovac@recipharm.com; 3Department of Medicinal Chemistry, Uppsala University, S-75123 Uppsala, Sweden; rafael.hartmann@ilk.uu.se; 4Oncopeptides AB, S-11153 Stockholm, Sweden; fredrik.lehmann@otpharma.se

**Keywords:** antibody–drug conjugate, sample preparation, mass spectrometry, desalting, trastuzumab, disulfide reduction

## Abstract

Antibody–drug conjugates (ADCs) are heterogeneous biotherapeutics and differ vastly in their physicochemical properties depending on their design. The number of small drug molecules covalently attached to each antibody molecule is commonly referred to as the drug-to-antibody ratio (DAR). Established analytical protocols for mass spectrometry (MS)-investigation of antibodies and ADCs often require sample treatment such as desalting or interchain disulfide bond reduction prior to analysis. Herein, the impact of the desalting and reduction steps—as well as the sample concentration and elapsed time between synthesis and analysis of DAR-values (as acquired by reversed phase liquid chromatography MS (RPLC–MS))—was investigated. It was found that the apparent DAR-values could fluctuate by up to 0.6 DAR units due to changes in the sample preparation workflow. For methods involving disulfide reduction by means of dithiothreitol (DTT), an acidic quench is recommended in order to increase DAR reliability. Furthermore, the addition of a desalting step was shown to benefit the ionization efficiencies in RPLC–MS. Finally, in the case of delayed analyses, samples can be stored at four degrees Celsius for up to one week but are better stored at −20 °C for longer periods of time. In conclusion, the results demonstrate that commonly used sample preparation procedures and storage conditions themselves may impact MS-derived DAR-values, which should be taken into account when evaluating analytical procedures.

## 1. Introduction

Antibody–drug conjugates (ADCs) are a class of biotherapeutics, which have gained clinical importance in oncology [1]. They are constructed by three main parts: an antibody backbone, a small molecule drug (payload) and a linker covalently connecting the two. Their construction encompasses a plethora of payloads, linkers, conjugation sites and monoclonal antibodies (mAbs) [2]. The conjugated payloads commonly affect both the chemical and physical properties, such as hydrophobicity and chemical or physical stability, compared to nonconjugated mAb [3]. Designing analytical protocols for ADCs entails many challenges due to their complexity, which surpasses that of many therapeutic proteins.

One of the most important parameters in ADC characterization is the drug-to-antibody ratio (DAR). The DAR-value quantifies the molar number of payloads attached to each antibody molecule and may impact the efficacy as well as the therapeutic window of a given ADC [4,5]. Many ADCs in development rely on native cysteine and lysine amino acid residues as conjugation sites. Due to the natural abundance of these amino acids, ADC synthesis commonly yields mixtures of conjugates with regard to both regiochemistry and DAR [6,7,8]. A DAR-value is therefore commonly reported as an average value of a given batch and directly correlates to the administrated dose of payload. The desired average DAR-value depends on the construct, but values around DAR 3–6 are most prevalent in those ADCs which have received marketing authorization [9,10,11]. ADCs with DAR-values ranging from one to eight or higher are also under development [12,13]. DAR-values can be assessed by many different techniques. Of these, hydrophobic interaction chromatography coupled to an ultraviolet and visual light detector (HIC–UV/Vis) and liquid chromatography–electrospray ionization mass spectrometry (LC–ESI–MS) are among the most common for ADCs conjugated via cysteine residues [14,15]. In recent years, multiple ESI–MS approaches for the characterization of mAbs and ADCs alike have been developed, using either native or denaturing ESI [15,16,17,18,19,20,21]. Native ESI–MS is appealing as it allows the detection of intact ADCs irrespective of the regiochemistry of payload attachment owing to the retention of the protein’s native structure in the gas phase [22]. However, native ESI–MS is particularly susceptible to interference from non-volatile salts in the samples as the presence of salts or impurities can lead to adduct formation or ion suppression [23]. As native-ESI may give rise to lower signals compared to denaturing ESI–MS [19], this study focuses on denaturing ESI–MS for DAR determination. However, the impact of a desalting step in sample preparation was still investigated.

This study was motivated by a lack of conclusions about the susceptibility of MS-derived DAR-values to alterations in sample treatment. There are several publications comparing HIC–UV/Vis to ESI–MS based approaches for DAR determination of cysteine-linked ADCs, although mainly for native ESI–MS [22,24,25]. As the trend seems to slowly shift from HIC–UV/Vis to ESI–MS as the main analytical technique of choice for ADC characterization [26,27], it is highly relevant to elucidate how susceptible ESI–MS derived DAR-values are to changes in the analytical protocol. A major difference between ESI–MS and HIC–UV/Vis analytical protocols lies in how the samples are treated. As the literature on the influence of the sample treatment and handling is limited, we have in this study endeavored to study the extent of its impact on ADC characterization, mainly focusing on the DAR determination.

In order to retain their 3D-structure, mAbs and ADCs are commonly stored in phosphate-buffered saline (PBS) or similar non-volatile salt mixtures [28]. In order to avoid ion suppression, it is necessary to remove these salts before native ESI–MS [29]. An array of desalting methodologies exist, but for most MS-analyses, it is common to use commercial size-exclusion gel filters or membrane devices, which separate analytes based on their molecular mass [30,31]. The cutoff mass can be chosen such that the protein is retained preferentially while salts and residues of free payload are removed. Moreover, these desalting techniques can have an effect on analyte concentration.

In denaturing ESI–MS it is often beneficial to include a reduction step prior to analysis in order to cleave any remaining disulfide bridges between the antibody’s heavy and light chains [15,32]. The reason for this is two-fold; (i) free light and heavy chains typically display higher sensitivity than intact ADCs as smaller molecules are easier to ionize and (ii) in cysteine-linked ADCs, some light and heavy chains are associated only by non-covalent forces. The gradually reduced hydrophilicity along the gradients deployed in reversed phase (RP) LC–ESI–MS disrupts the non-covalent forces between light and heavy chains [2,19]. Consequentially, unless it is fully reduced prior to analysis, a sample thus consists of a mixture of free light and heavy chain species and intact ADC.

This study is based on the sample treatment workflows commonly used in publications for DAR determination using RPLC–ESI–MS and native ESI–MS (see Figure 1). The following common steps were given particular emphasis in this study: interchain disulfide reduction and desalting/buffer exchange. Additionally, sample concentration and the impact of delaying the analysis were also briefly investigated.

## 2. Materials and Methods

### 2.1. Chemicals

*N*-glycosidase F from *Elizabethkungia meningoseptica* (PNGase F, #F8435-50UN, CAS 83,534–39–8), dithiothreitol (DTT), iodoacetamide (IAA), sodium chloride, acetonitrile and formic acid were purchased from Sigma-Aldrich, MO, USA. Ammonium acetate and isopropanol were purchased from Carlo Erba, Spain. Dibasic sodium phosphate was purchased from Merck, Germany. Maleimidocaproyl-valinyl-citrullinyl *para*-aminobenzyloxycarbonyl monomethyl auristatin E (Mc–VC–PABC–MMAE) (abbreviated vcMMAE, # AB456059, CAS 646502-53-6) was purchased from abcr GmbH, Germany and *N*-succinimidyl 4-(maleimidomethyl)cyclohexanecarboxylate (SMCC)-DM1 (abbreviated DM1, # HY-101070, CAS 1228105–51–8 was purchased from MedChemExpress, NJ, USA. MS-grade solutions and water purified with a lab purification system (Millipore) were used for the preparation of mobile phases and sample dilutions.

### 2.2. Antibody and ADC Samples

Trastuzumab was purchased in PBS solution (20 mg/mL) from Carbosynth Ltd. (#FT65040, UK) and vcMMAE- and DM1-conjugates thereof were synthesized in-house in single batches. VcMMAE–trastuzumab was obtained by the reduction of endogenous cysteine disulfide bridges of the antibody, followed by subsequent Michael addition using the same procedure as published by J. Francisco et al. [33]. DM1–trastuzumab was synthesized by conjugation to endogenous lysine residues of trastuzumab in an analogous fashion as reported by Hamann et al. [34].

### 2.3. Deglycosylation and Disulfide Reduction

Two aliquots of 400–600 µg trastuzumab/ADC were diluted to 1 mg/mL and treated with PNGaseF (500 units/µL, 1 µL to 25 µg protein) at 37 °C (500 units/µL) for 24 h. One aliquot was then reduced by adding 1 µL 1-M DTT for 10 µL of sample–solution and incubated at room temperature for 30 min. A small aliquot of 20–30 µL sample–solution was taken out and the rest of the samples were kept at 4 °C until desalted (as described below).

Additional sets of samples were deglycosylated in the same manner as described above. Twenty microliters of each vcMMAE–trastuzumab and nonconjugated trastuzumab were reduced using 1-M DTT (1:10, V:V) for 30 min and further treated by adding 0.5 µL 550-mM IAA to the samples and incubating them in a dark location at room temperature for 20 min before analysis.

An aliquot from the previous deglycosylated samples (trastuzumab/ADC) was reduced in the same manner as described above, but the reduction was left to progress for different lengths of time. The samples were mixed with 1-M DTT (1:10) and left at room temperature. Ten microliters of the sample were taken out at set time points between 30 min up to 4 h and mixed 1:1 with a solution containing 15% acetonitrile and 0.5% formic acid to halt the reduction.

### 2.4. Desalting

#### 2.4.1. Spintrap

One hundred microliters of deglycosylated ADCs and mAb (1 mg/mL in PBS) were transferred (in duplicates) to GE Spintrap PD 25 spin columns (GE Healthcare, #28-9180-04, UK) that was preconditioned with 100-mM ammonium acetate solution. The samples were eluted according to the protocol provided by the manufacturers using 100-mM ammonium acetate for the buffer exchange. The eluates were collected in clean sample tubes that was weighed prior and after elution to estimate the sample volume of the eluates. The samples were transferred to glass vials fitted with 250 µL inserts and kept at 4 °C until further processing. Finally, formic acid and acetonitrile were added to 20-µL of each sample, to a final concentration of 0.5% formic acid and 20% acetonitrile.

One hundred microliters of deglycosylated and reduced ADCs and mAb (1 mg/mL in PBS) were desalted in the same fashion but using 0.5% formic acid to prewash the columns and elute the samples. Finally, 10 µL acetonitrile was added to each of the reduced samples.

#### 2.4.2. Spin Filter

One hundred microliters of the deglycosylated ADCs and mAb (1 mg/mL in PBS) were transferred (in duplicates) into Amicon Ultra 0.5 mL 30-kDa spin filters (Merck, #UFC503024, Germany) prewashed with 100-mM ammonium acetate (centrifuged at 10,000 rpm until almost complete dryness). One hundred and fifty microliters of 100-mM ammonium acetate solution was then added to the sample solution in each filter before centrifugation for 10 min at 10,000 rpm. To elute the sample, the filters were inverted into clean and weighed collection tubes and centrifuged at 4000 rpm for 2 min. The collection tubes with filters were weighed before and after elution to estimate the sample volume of the eluates. Finally, the samples were transferred into glass vials fitted with 250 µL inserts and 100-mM ammonium acetate solution was added to each sample to obtain a final volume of 100 µL before placing the samples in 4 °C until further processing. Finally, formic acid and acetonitrile were added to 20 µL of each sample, to a final concentration of 0.5% formic acid and 20% acetonitrile.

One hundred microliters of deglycosylated and reduced ADCs and mAb (1 mg/mL in PBS) were desalted in the same manner as above. However, Amicon Ultra 0.5 mL 3-kDa spin filters (Merck, #UFC500324, Germany) prewashed with 0.5% formic acid was used instead. One hundred and fifty microliters of 0.5% formic acid in 15% acetonitrile was added to each sample before the initial centrifugation (30 min at 10,000 rpm) and a solution containing 0.5% formic acid in 15% acetonitrile was used to dilute the final volume of each sample to approximately 100 µL.

### 2.5. Dilution Series

Three dilution series were made from desalted trastuzumab and ADCs and reference samples (with no desalting step included in the workflow). For each dilution series, a portion of each deglycosylated sample was taken out and reduced as described above (mixing 1:10 with 1-M DTT and incubating for 30 min). The samples were mixed 1:1 with 0.5% formic acid in 15% acetonitrile and a portion of the sample was taken out and diluted in consecutive dilutions (1:1, 1:2 and 1:4) with 0.5% formic acid in 15% acetonitrile resulting in four solutions with decreasing concentration.

### 2.6. RPLC–MS

All the above-mentioned samples were analyzed using a Waters Xevo TQ-S Micro triple quadrupole MS equipped with a Z-spray ESI source connected to a Waters Acquity UPLC system. A Waters Acquity UPLC Protein BEH C4 column 2.1 mm × 50 mm with pre-column was used in combination with mobile phases consisting of 0.5% formic acid (A) and 0.5% formic acid in acetonitrile (B). After an initial isocratic step of 25% mobile phase B for 1 min, a gradient from 25%–85% mobile phase B was applied over 8 min. The total run time was 10 min, which included a 2 min re-equilibration step at the end. The flow rate was kept at 0.4 mL/min and the column temperature was set to 50 °C. Ten microliters of trastuzumab and ADC samples were injected in triplicates after being diluted 1:1 with formic acid in acetonitrile to a final concentration of 0.5% formic acid and 15% acetonitrile just before analysis. Spectra were collected in MS scan mode from 2–10 min into each run over the mass range 500–2040 *m/z* calibrated at 0.75 *m/z* FWHM unit resolution. The source temperature was set to 150 °C, desolvation gas flow rate was 1200 L/h, the capillary voltage was 3.0 kV, cove voltage was set to 40 V and the desolvation temperature was set to 650 °C.

### 2.7. Storage Samples

For storage tests (analyzed by size exclusion chromatography(SEC)-UV/Vis), trastuzumab and both ADCs (in PBS) were diluted to a final concentration of 1 mg/mL with water and mixed properly and aliquoted into Agilent’s glass vials with 250-µL glass insert with 35 µL of sample–solution in each vial. The vials were then sealed with parafilm. Four vials were stored at 4 °C for 1 week to 41 days and another four vials were frozen by immersion into ethanol and dry ice mixture before storage at −20 °C for 1 week to 44 days.

For monitoring DAR changes over time, samples were taken from the original batches upon conjugation, after an one-month storage at 4 °C (upon which point the original batch was frozen) and after an additional three months storage at −20 °C.

### 2.8. SEC–UV/Vis

Ten microliters of sample was injected in triplicate and run on a Waters UPLC SEC BEH 300 Å 2.1 × 150-mm column with pre-column using an Agilent 1100 LC-system with a diode array detection (DAD) detector. A 15-minute isocratic elution was done with 200-mM sodium chloride, 15% isopropanol and 100-mM dibasic sodium phosphate at (pH 8) at a flow rate of 0.3 mL/min. The absorbance signal was monitored at 220 and 280-nm wavelengths for all samples.

### 2.9. Data Analysis

All spectra were deconvoluted for both light and heavy chains simultaneously using Waters’ MassLynx software with the MaxEnt1 add-on over the mass range of 23–56 kDa. The resolution was set to 1.0 Da/channel and width at half height was 1.0 Da. Minimum intensity ratios of 50% (left) and 60% (right) and a maximum of 20 iterations were used. Deconvolution was made from spectra extracted over the same timeframe for each run of one particular ADC/mAb, irrespective of sample treatment. Any observed changes were examined for significance by one-way ANOVA using the function in Microsoft Excel. To check for significant differences between the two desalting options on the acquired DAR-values, a two-way ANOVA with replicates was used comparing the duplicate samples. Average DAR-values are done using the deconvoluted peak area from mass spectra containing free light and heavy chains of distinct conjugation degree. The calculations are based on equations reported elsewhere [35] with the weight percent initially calculated separately for light chain species and heavy chains species before summarized to an overall average DAR-value for each ADC batch.

## 3. Results

Two ADCs were synthesized in-house by conjugation of trastuzumab to the two hydrophobic payloads MMAE and DM1, respectively. The ADCs were based on the same antibody in order to allow for differences in behavior during sample preparation originating from the conjugation chemistry to be more readily observed. Consideration was also taken with regard to the payloads’ prevalence in literature. The MMAE payload was conjugated to endogenous cysteines obtained by disulfide reduction and subsequent Michael addition with a peptidase cleavable linker, while the DM1 payload was conjugated to endogenous lysines via a non-cleavable linker.

### 3.1. The Effect of the Time Frame of the Interchain Disulfide Reduction on acquired DAR-Values

DTT is a common reducing agent, which, when used in excess at basic pH, typically reduces all protein–disulfides within thirty minutes [36]. To prevent the reducing agent from further reaction, minimize unwanted degradation of the analyte and prevent reformation of the disulfide bridges, an alkylation agent can be added, e.g., iodoacetamide [37]. The alkylation was found to give rise to additional peaks that were found to be most pronounced on the heavy chains of cysteine-linked ADCs (see Appendix A and is therefore commonly not used for DAR characterization. Instead, in order to prevent thiol reoxidation, the reducing agent is often quenched prior to injection by the addition of an acid [1,15,38,39]. However, the exact timing of the acidification is only described sporadically in literature and is completely excluded in some workflows. This study endeavored to elucidate the impact of the time elapsed between the addition of reductant and the acidic quench on the final DAR-value. Incremental reaction times from 30 min to 4 h were assessed.

The DAR was calculated from samples that had been incubated with DTT for 30 min, 60 min, 2 h or 4 h prior to the addition of formic acid. These analyses were performed on all samples of mAb and ADC alike. For vcMMAE–trastuzumab, no differences were visible in the acquired mass spectra for samples that had been acidified after more than the usual 30 min. However, for DM1–trastuzumab and unconjugated trastuzumab, a shift towards lower mass-to-charge values could be observed in the charge envelopes of the light and heavy chains (see Appendix A). The difference was pronounced after 60 min (see Figure 2) but changed very little when prolonging the reduction time even further for DM1–trastuzumab. For nonconjugated trastuzumab, the shift progressed over the entire period of observation.

The apparent DAR-values decreased for both ADCs as reduction periods were prolonged (see Table 1). Given the minor differences in the mass spectra of vcMMAE–trastuzumab, the effect on DAR was not significant, as confirmed by one-way ANOVA statistical testing (data not shown). On the other hand, the decrease in DAR-value observed for DM1–trastuzumab was shown to be significant (*p* < 0.05), which also conforms well with the aforementioned changes in the mass spectra. Whether the different levels of susceptibility of the DAR-values of the ADCs originate from differences in regiochemistry, payload or average DAR-value has yet to be examined.

### 3.2. The Effect of Adding a Desalting Step on the Acquired DAR-Values

For comparison of desalting approaches, the focus in this study has been on desalting options suitable for sample volumes of 100 µL or less. Specifically, one ultracentrifugation filtering device (Amicon Ultra 0.5-mL spin filter) and one centrifugation gel–filtration device (GE Spintrap PD 25 spin columns) were examined, as they are commonly deployed for the desalting of mAbs and ADCs [24,25,40,41]. For a rapid and robust analysis, RPLC–ESI–MS was used instead of native ESI–MS.

As the aforementioned devices have been designed for the desalting of endogenous proteins, it was deemed relevant to ascertain that the recovery of ADC samples did not differ significantly from that of nonconjugated antibody.

A comparison of the peak areas in the total ion chromatograms of samples desalted by either spintraps or spin filters to a reference sample, made in parallel to the other samples, but without desalting, revealed that neither of the tested desalting devices led to significant loss of sample for either mAb or ADCs. Recoveries were found to be close to 100% or higher by RPLC–MS. However, spintraps yielded a lower recovery than the spin filters for all samples (see Appendix A). Recoveries above 100% may be attributed to the improved ionization efficiency associated with the replacement of non-volatile salts in PBS with ammonium acetate prior to injection, resulting in a decrease of ion suppression and a consequent increase of the TIC. It should, therefore, be noted that recoveries derived from MS-data must also be considered subject to differences in ionization [41], as well as the more widely known adsorption related losses.

Bradford protein quantification assay gave recoveries of 80% or higher, but confirmed the relative change observed in RPLC–MS (see Appendix A). Therefore, no significant loss of sample occurred as a result of either desalting protocol for the ADCs tested in this study.

### 3.3. Ionization

As the DAR-value is calculated using relative intensities of the different DAR species in the sample mixture, it relies on all conjugation species having the same or similar, ionization efficiency [2]. However, as was shown in previous studies, this is not always the case [27,41]. A slight shift in the charge envelope towards lower *m/z* could be observed for the desalted mAb/ADCs compared to mass spectra obtained from reference samples (see Appendix A). The mass shift was most pronounced for DM1–trastuzumab.

As can be observed in Figure 3, the effect of the desalting step on the apparent DAR-values of both ADCs depended on the timing of the desalting step in relation to the reduction step. If desalting was done prior to the disulfide reduction step, the apparent DAR-value for DM1–trastuzumab decreased slightly. When reduction preceded desalting, the DAR-value was seen to increase. The change was similar irrespective of which desalting device was used, if done prior to reduction, but was found to be slightly larger if done using spintrap devices for already reduced samples. On the other hand, for vcMMAE–trastuzumab, the relationship between how the apparent DAR-value differed (compared to each reference sample) depending on the timing of the desalting step was reversed. Moreover, the spintrap impacted the DAR-value slightly more for vcMMAE–trastuzumab. All changes were found to be significant (*p* < 0.05) in both cases by one-way ANOVA analyses.

The DM1–trastuzumab remains intact no matter the degree of conjugation until reduced, while vcMMAE–trastuzumab may exist in both free and covalently linked light and heavy chains depending on the degree of conjugation. Thus, potential losses due to differences in adsorption between conjugation species in the desalting device could potentially impact the two types of ADCs differently at the reduced or intact level.

### 3.4. The Effect of Sample Concentration on the DAR-Value Acquisition

The ideal sample concentration in ESI–MS depends on the choice of instrument and ESI source. Many publications use sample concentrations of ≥1 µg/µL for optimal signal intensity in RPLC–ESI–MS [7,8]. However, in native ESI–MS it can be beneficial to work with lower analyte concentrations for optimal desolvation. As the relative intensity of different conjugations species in a sample batch can differ significantly, two ADCs with different average DAR-values were used to test if the sample concentration could impact the acquired DAR-values for a given batch.

Dilution of the samples prior to injection had a different impact on the apparent DAR-values depending on the extent of the observed changes in relative peak intensities in the deconvoluted spectra of a particular ADC. The DM1–ADC used in this study had a lower DAR than the vcMMAE–conjugate. If the highest conjugation species are close to the limit of detection as in this case (see Appendix A) dilution will naturally lower the DAR. As can be observed in Figure 4, for DM1–trastuzumab, the apparent average DAR-value decreased upon dilution for both desalted and reference samples. One-way ANOVA analyses found all observed changes to be significant (*p* < 0.05) for DM1–trastuzumab.

If no conjugation species was close to the limit of detection as in the case of vcMMAE–trastuzumab (see Appendix A), the apparent DAR was either unaffected (for spintrap desalted samples and the reference sample) or increased slightly (for samples desalted by spin-filtration) when the samples were diluted (see Figure 5). For vcMMAE–trastuzumab, only the difference observed upon dilution of the sample desalted by a spin filter had a significant impact on apparent DAR (*p* < 0.05, one-way ANOVA).

### 3.5. Storage Conditions for ADC Samples

Protein samples are known to be sensitive to their storage conditions [42]. To prevent decomposition of the samples, they are generally stored at 4 °C in liquid solutions or, more commonly, frozen at −20 °C or −80 °C. mAbs and ADCs are susceptible to aggregation when exposed to repeated freeze cycles and are therefore often stored at 2–8 °C [43]. An aim of this study was to investigate the impact of storage temperature on aggregation and fragmentation and to thus arrive at a recommendation for storage conditions in the case of delayed analyses. The degree of aggregation and fragmentation was monitored by size exclusion chromatography (SEC–UV/Vis), while the effect on apparent DAR was examined by means of RPLC–ESI–MS.

The storage of samples under freezing conditions generally results in a less pronounced increase in fragmentation compared to storage at 4 °C for both nonconjugated trastuzumab and both ADCs over the course of 41–44 days (see Appendix A). However, DM1–trastuzumab showed only minor changes in fragmentation over time, while fragmentation was more pronounced for vcMMAE, irrespective of storage temperature. No clear trend could be observed for sample aggregation other than that the variation upon sample storage did not exceed 2% for either mAb or the ADCs (see Appendix A).

In parallel to the extension of aggregation in the samples, it is interesting to investigate if the DAR-value may also be impacted in case of unexpected delays in the analysis. As can be observed in Figure 6, storing the ADCs for one month at 4 °C prior to DAR determination lead to minor, but statistically significant (as tested with one-way ANOVA, *p* < 0.05) decrease in the apparent DAR-value of both ADCs, similar to samples exposed to longer reduction times.. As the SEC-data confirmed that −20 °C showed less fragmentation of both ADC samples, after one month the samples were frozen for any additional re-analyses. To our surprise, the additional storage of 3 months at −20 °C led to a slight, but significant increase in the apparent DAR-values for both ADCs.

## 4. Discussion

The impact of two different sample preparation steps (disulfide reduction and desalting/buffer exchange) in ADC sample treatment prior to ESI–MS DAR determination was investigated in this study. Two different ADCs based on trastuzumab (selected due to its extensive characterization in literature and its use as a standalone drug) were synthesized using two different payloads and served as analytes.

The prolongation of the disulfide reduction reaction prior to the analysis was shown to impact the apparent DAR-value of both types of ADCs to some degree. Longer reduction times affected the charge envelope of both light and heavy chains for mAb and DM1- trastuzumab. A shift in the charge envelope for proteins in denaturing ESI–MS to higher charge states often correlates with an unfolding of the protein structure [44]. Furthermore, prolonged reduction can lead to complete disulfide reduction, which would also open up the protein structure, exposing more proton binding sites [45]. As the antibody had already been exposed to reducing agent during the synthesis of the vcMMAE–trastuzumab, it can be speculated that the 3D-structure of that ADC was most likely already more readily solvent-accessible before the addition of the reducing agent in the sample treatment. As a result, prolonged exposure to DTT could have had less of an impact on the charge envelope of vcMMAE–trastuzumab than it did in the case of DM1–trastuzumab and nonconjugated trastuzumab. However, as a larger shift in the charge envelope (see Appendix A) correlated with a larger shift in the apparent DAR-value (see Table 1) there is indication that charge states directly impact the apparent DAR-values. Considering the fluctuations of DAR-values observed in this study for prolonged reduction time, it can be concluded that when ADC samples are kept in the presence of DTT to prevent reoxidation prior to and during analysis, a small amount of acid should always be added after 30 minutes of reaction time to quench any further reduction.

For both tested desalting protocols, recoveries above 80% were confirmed by Bradford for both investigated ADCs, despite their increased hydrophobicity (than nonconjugated trastuzumab, see Appendix A). However, it should be noted that the synthesis of the ADCs included a gel–filtration step (similar to the spintrap desalting step) after payload conjugation. Thus, it is possible that bigger discrepancies in recovery may have been observed had this step been excluded. Although the recoveries of both ADCs were similar, it could also be concluded that the desalting method affected the apparent DAR of the two conjugates (see Figure 3). The observed changes were found to depend on when the desalting was performed in relation to the reduction step. However, the impact on the DAR-values was within ±0.3 of the DAR determined for the reference samples. Although on-column desalting is a de facto effect of RPLC, Lazar et al. have found indications that RPLC alone did not remove all salt adducts prior to ESI–MS [12]. The increase in peak area in the TIC for all desalted samples in this study further confirmed that offline desalting may be beneficial for RPLC–MS-analyses for enhanced salt removal.

As alterations in the sample preparation workflows can affect the final protein concentration prior to injection, the impact of sample dilution on the apparent DAR was also taken into consideration. As anticipated, ADCs with DAR species of low intensity are more susceptible to decreases in concentration, which may lead to a principal underestimation of average DAR. This is exactly what was observed for the DM1–trastuzumab analyte (see Figure 4), which had an average DAR-value of 1.5. Contrary, if all conjugation species had a clearly visible peak in the mass spectra, the sample concentration was less consequential (as observed for vcMMAE–trastuzumab in Appendix A). However, to our surprise, if the sample had been desalted using spin filters prior to dilution; vcMMAE–trastuzumab samples yielded higher apparent DAR-values upon dilution (see Figure 5). A possible explanation for this observation is that any salt residues will also be diluted alongside the ADC so that suppression effects on higher conjugation species decrease. It is thus recommended to determine DAR-values at two different sample concentrations, when possible.

Last, as samples may sometimes require unforeseen storage before analysis can be performed, the effect of storage conditions on the apparent DAR-value was examined. At the two temperatures tested (4 °C and −20 °C), the lysine-linked ADC investigated in this study proved to be relatively insensitive to prolonged storage, while the cysteine-linked ADC proved to benefit from being kept frozen (−20 °C) if analyses are to be delayed (see Appendix A). Furthermore, the acquired data indicated that freezing ADCs upon storage likely prevents deconjugation. In conclusion, although no significant change in aggregation was observed for either ADC, storing the samples at 4 °C for up to a month was shown to lead to a gradual decrease in the apparent DAR-values. Overall, it could be concluded that for the ADCs in the study freezing the samples at −20 °C better retained the integrity of the sample for delayed analyses/re-analyses.

As ADCs are inherently heterogeneous analytes, the findings of this study may differ for other constructs, different average DAR-values and depending on the instrument used for analysis. However, it can be concluded that DAR-values determined by mass spectrometry are susceptible to changes in sample treatment (albeit small). Furthermore, altering the antibody used in the ADCs can impact both the DAR-value and physio-chemical properties such as overall hydrophobicity and temperature stability [46]. Hydrophobicity can have a huge impact on aggregation tendencies and potential sample losses [47], especially during desalting and the selected sample preparation protocol needs to be validated not only for a given DAR range, but also for each mAb and linker–drug combination.

## Figures and Tables

**Figure 1 antibodies-09-00046-f001:**
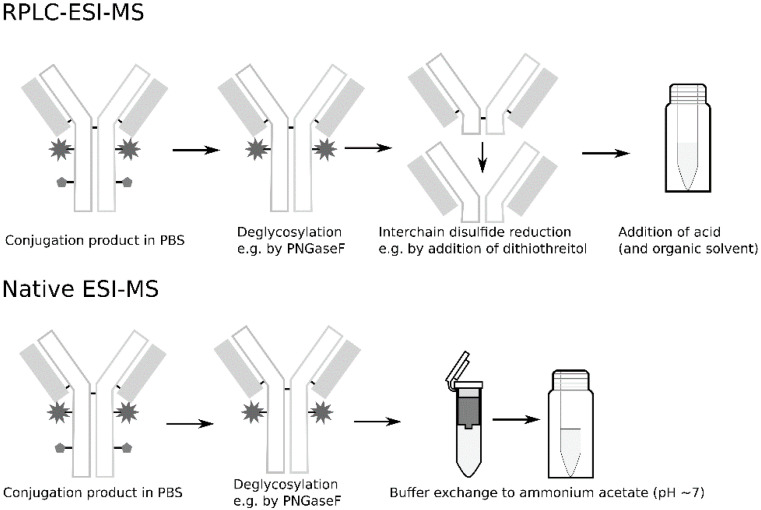
Generic workflows of sample treatment for denaturing and native electrospray ionization mass spectrometry (ESI–MS) protocols used for characterizations of monoclonal antibodies (mAbs) and drug-to-antibody ratio (DAR) determination of antibody–drug conjugates (ADCs).

**Figure 2 antibodies-09-00046-f002:**
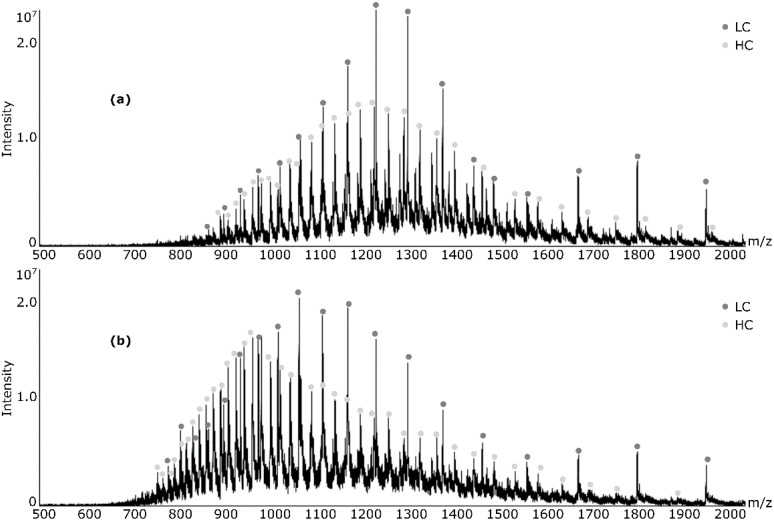
Observed shift in charge envelope towards lower mass-to-charge values for *N*-succinimidyl 4-(maleimidomethyl)cyclohexanecarboxylate (SMCC)-DM1 (DM1)–trastuzumab when subjected to disulfide reduction for (**a**) 30 minutes and (**b**) 60 minutes.

**Figure 3 antibodies-09-00046-f003:**
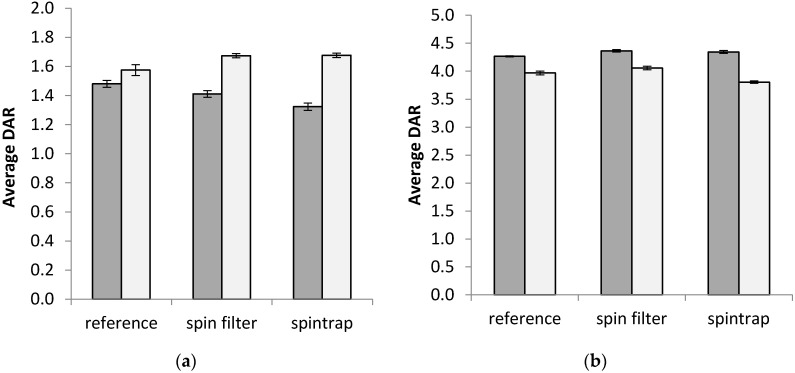
Average apparent DAR-values of ADCs depending on desalting method. Depending on the desalting method, the average DAR showed different behavior in relation to if the desalting was done on intact or free light and heavy chains for the two investigated ADCs (**a**) vcMMAE–trastuzumab and (**b**) DM1–trastuzumab. Gray bars—desalting prior reduction, White bars—desalting after reduction. As desalting prior to or after the reduction step was prepared on two different dates, a fresh reference without offline desalting was made for each sample set. Spintrap (GE PD spintrap gel–filtration columns) and spin filter (Amicon Ultra 3 kDa, after or 30 kDa, before, MWCO) were prepared in duplicates and reported values are combined values of triplicate injections over both replicates. Error bars depict the standard deviation from triplicate measurements for each sample (for two samples combined for the two desalting options).

**Figure 4 antibodies-09-00046-f004:**
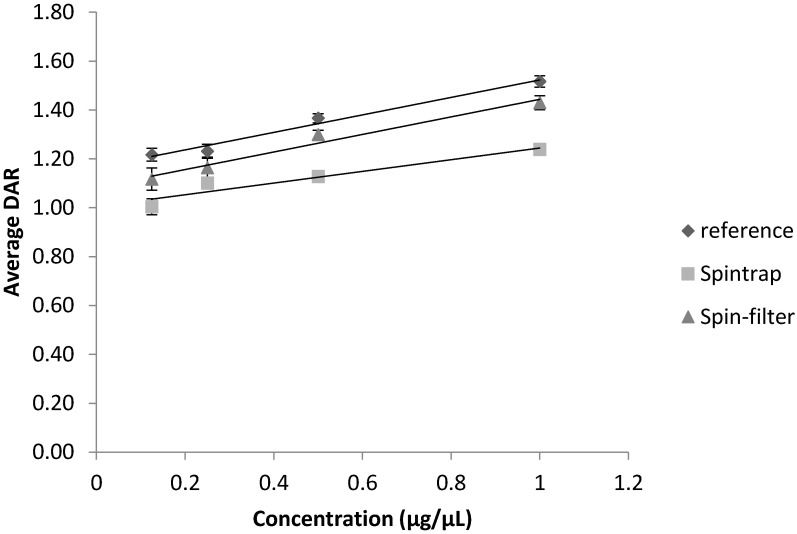
Apparent DAR-values of DM1–trastuzumab depending on sample concentration prior to injection and effect of sample desalting Error bars depict the standard deviation from triplicate measurements for each sample.

**Figure 5 antibodies-09-00046-f005:**
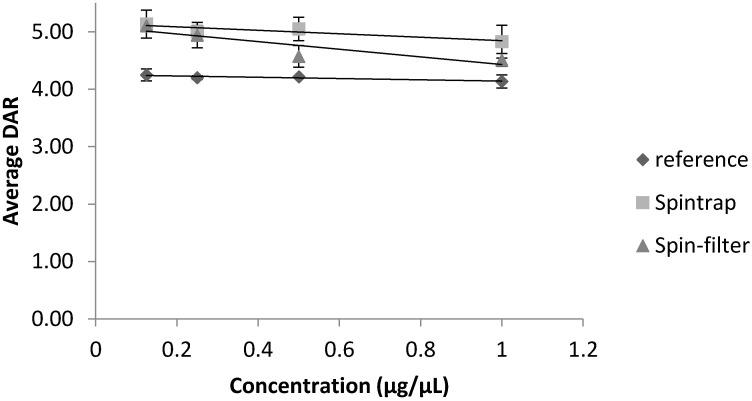
Apparent DAR-values of vcMMAE–trastuzumab depending on sample concentration prior to injection and effect of sample desalting. Error bars depict the standard deviation from triplicate measurements for each sample.

**Figure 6 antibodies-09-00046-f006:**
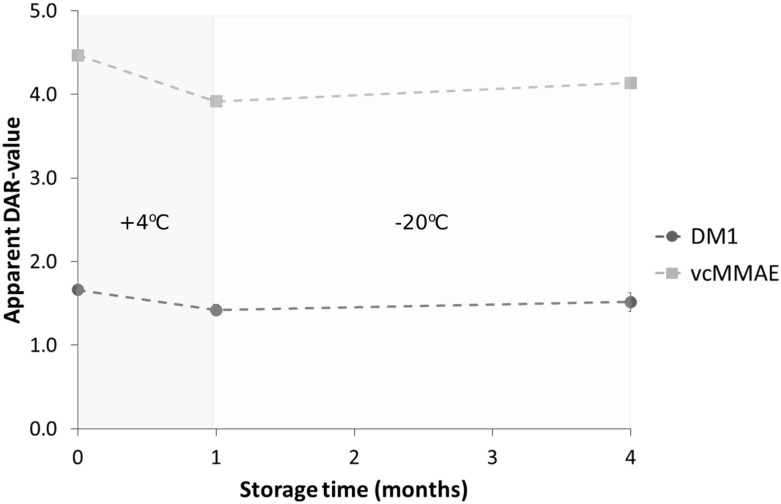
Dependence of apparent DAR-values on storage of ADC samples in case of delayed analyses. The apparent DAR-values for one single batch of either cysteine-linked (vcMMAE) and lysine-linked (DM1) ADC, as determined by RPLC–MS, changed over time from storage at 4 °C for 1 month (decrease in apparent DAR-values) followed by −20 °C storage for an additional 4 months (increase in apparent DAR-values). All samples were stored in phosphate-buffered saline (PBS) solution at 1-µg/µL protein concentration. Error bars depict the standard deviation from triplicate measurements for each sample.

**Table 1 antibodies-09-00046-t001:** Average DAR-values of ADCs subjected to reduction for different incubation times.

ADC	30 min ^1^	60 min ^1^	2 h ^1^	4 h ^1^
vcMMAE–trastuzumab	4.3 ± 0.10	4.2 ± 0.17	4.2 ± 0.02	4.1 ± 0.19
DM1–trastuzumab	0.9 ± 0.02	0.7 ± 0.10	0.6 ± 0.05	0.6 ± 0.01

^1^ Data acquired by RPLC–MS from triplicate injections. ADCs, Antibody–drug conjugates.

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
