# Peer review of "Investigating the Impact of Sample Preparation on Mass Spectrometry-Based Drug-To-Antibody Ratio Determination for Cysteine- and Lysine-Linked Antibody–Drug Conjugates"

_2073-4468, 2020, doi:10.3390/antib9030046_

Round 1

Reviewer 1 Report

This paper is an excellent methodology of sample preparation on mass spectrometry analysis for ADCs, which is very clear and to the point. But I would like the author to revise the following points.

  1. I do not understand the meaning of the experimental conditions (storage at 4°C for 1 month followed by -20°C storage for an additional 4 months) in Figure 6. The author should clarify the purpose of the experimental condition. The author should add error bars in Figure 6.
  1. The author should discuss the most optimal condition for sample preparation of each ADC in the Discussion.

Author Response

This paper is an excellent methodology of sample preparation on mass spectrometry analysis for ADCs, which is very clear and to the point. But I would like the author to revise the following points.

  1. I do not understand the meaning of the experimental conditions (storage at 4°C for 1 month followed by -20°C storage for an additional 4 months) in Figure 6. The author should clarify the purpose of the experimental condition. The author should add error bars in Figure 6.

Error bars have been added to Figure 6, although the error is so small that for most data points the error bars are covered by the data point. The intention of the experimental conditions behind this data has been described in more detail on line 404-412.

  1. The author should discuss the most optimal condition for sample preparation of each ADC in the Discussion.

Recommendations are stated in the text on the following lines: 433-436, 448-450, 461-462 and 471-472. As it is not possible to know the true DAR-value for an ADC (as all techniques has some kind of inherent bias) it is not possible to do a final recommendation on which option is best in regards for DAR determination (as it is not possible to say which changes bring the value closer or further away from the true value). The overall conclusion instead lies in the fact that it is important to be aware of these biases when doing DAR determinations with MS-based methodologies, which is discussed on line 478-483.

Reviewer 2 Report

This paper by Kaellsten et al is a solid piece of work describing antibody preparation for in vitro characterisation to determine the DAR value via MS. A lot of work went into this work.

The manuscript is well written and logical.

Please explain ALL abbreviation especially in the abstract (e.g. DTT is explained at line 231)

Please confirm all your statements with either data sets or literature. The Introduction and Discussion are missing literature (e.g. line 51).

It would interesting to know why HER2 mAb was used for this study. To confirm the data sets, a second antibody would have been very interesting. It is known that the DAR-values are differing between various antibodies – please discuss what that would mean for your results.

Figure 6 – please explain the choice of time points vs temperature. It sound unusual to keep the antibody first for one month at 4 C before freezing it at -20C.

Author Response

This paper by Kaellsten et al is a solid piece of work describing antibody preparation for in vitro characterisation to determine the DAR value via MS. A lot of work went into this work.

The manuscript is well written and logical.

Please explain ALL abbreviation especially in the abstract (e.g. DTT is explained at line 231)

The text has been overhauled and any abbreviations without definition in the abstract and the text have been corrected (see tracked changes).

Please confirm all your statements with either data sets or literature. The Introduction and Discussion are missing literature (e.g. line 51).

References to papers that support the statement on line 58 (previously line 51) have been added. References have also been added on line 73, 82, 89, 102, 309, 382, 385, 423, 425, 485 and 486 and referral to data on line 430-431, 439, 443, 455, 457-458, 460 and 467 in the discussion for increased clarity.

It would interesting to know why HER2 mAb was used for this study. To confirm the data sets, a second antibody would have been very interesting. It is known that the DAR-values are differing between various antibodies – please discuss what that would mean for your results.

The motivation for selecting HER2 and some considerations that would have to be made in case the given protocols are to be transferred to the analysis of ADCs based on a different antibody have been discussed on line 416-417 (motivation) and 483-488(practical importance).

Figure 6 – please explain the choice of time points vs temperature. It sound unusual to keep the antibody first for one month at 4 C before freezing it at -20C.

As mentioned above, the reasoning behind the experimental conditions behind this data has been described in more detail on line 404-412.
